# The ANGPTL3-4-8 Axis in Normal Gestation and in Gestational Diabetes, and Its Potential Involvement in Fetal Growth

**DOI:** 10.3390/ijms24032486

**Published:** 2023-01-27

**Authors:** Sergiy Klid, Elsa Maymó-Masip, Francisco Algaba-Chueca, Mónica Ballesteros, Montserrat Inglès-Puig, Albert Guarque, Ana Madeira, Carlos Jareño, Joan Vendrell, Sonia Fernández-Veledo, Ana Megía

**Affiliations:** 1Department of Medicine and Surgery, Rovira i Virgili University, 43005 Tarragona, Spain; 2CIBER de Diabetes y Enfermedades Metabólicas Asociadas (CIBERDEM), Carlos III Health Institute, 28029 Madrid, Spain; 3Departament of Endocrinology and Nutrition, Research Unit, University Hospital of Tarragona Joan XXIII, Institut d’Investigació Sanitària Pere Virgili (IISPV), 43007 Tarragona, Spain; 4Department of Obstetrics and Gynecology, University Hospital of Tarragona Joan XXIII, Institut d’Investigació Sanitària Pere Virgili (IISPV), 43007 Tarragona, Spain

**Keywords:** gestational diabetes, ANGPTL, fetal growth, lipid metabolism

## Abstract

Dyslipidemia in gestational diabetes has been associated with worse perinatal outcomes. The ANGPTL3-4-8 axis regulates lipid metabolism, especially in the transition from fasting to feeding. In this study, we evaluated the response of ANGPTL3, 4, and 8 after the intake of a mixed meal in women with normal glucose tolerance and gestational diabetes, and we assessed their gene expressions in different placental locations. Regarding the circulating levels of ANGPTL3, 4, and 8, we observed an absence of ANGPTL4 response after the intake of the meal in the GDM group compared to its presence in the control group. At the placental level, we observed a glucose tolerance-dependent expression pattern of *ANGPTL3* between the two placental sides. When we compared the GDM pregnancies with the control pregnancies, a downregulation of the maternal side *ANGPTL3* expression was observed. This suggests a dysregulation of the ANGPTL3-4-8 axis in GDM, both at the circulating level after ingestion and at the level of placental expression. Furthermore, we discerned that the expressions of *ANGPTL3*, *4*, and *8* were related to birth weight and placental weight in the GDM group, but not in the control group, which suggests that they may play a role in regulating the transplacental passage of nutrients.

## 1. Introduction

Angiopoietin-like (ANGPTL) proteins are a family of secreted proteins with eight known members, ANGPTL1-8. ANGPTL3, ANGPTL4, and ANGPTL8 are three members of this family with pleiotropic functions, which include the regulation of plasma triglyceride levels, as well as of several physiological functions such as stem cell renewal, angiogenesis, and inflammation [1,2,3,4]. They are mainly expressed and secreted by the liver and adipose tissue, but they can also be expressed in other tissues, including the placenta [5,6,7]. They selectively inhibit the lipoprotein lipase (LPL) enzyme activity in different metabolic compartments in a coordinated way to maintain metabolic homeostasis. The interplay between ANGPTL3, 4, and 8 seems to be a crucial regulator of triglycerides trafficking during feeding and fasting [8,9,10].

A normal pregnancy is characterized by increased lipid concentrations, especially triglycerides, which provide fuel and nutrients for both the placenta and the developing fetus [11]. In addition to glucose and amino acids, lipid transfer through the placenta is essential for fetal growth and development [12]. Early gestation is characterized by an increase in LPL activity, which favors the hydrolysis of circulating triglycerides and facilitates the accumulation of lipids in maternal depots. During late pregnancy, the requirements for metabolic substrates are augmented. A decrease in insulin sensitivity and an increased lipolytic activity in the maternal adipose tissue, as well as attenuated lipid uptake by the peripheral tissues, results in a physiological maternal hyperlipidemia.

Both maternal obesity and gestational diabetes mellitus (GDM) worsen the lipid maternal profile, and these changes are associated with obstetrical and perinatal complications. In pregnancies complicated with diabetes, maternal dyslipidemia seems to enhance the availability of lipids to the fetus and contribute to fetal overgrowth. Even in well-controlled GDM, maternal triglycerides and free fatty acids are positively correlated with neonatal weight and fat mass [13,14]. It has been proposed that although lipids cross the placenta with difficulty, changes in the placental function in the context of hyperlipidemia associated with GDM could lead to an increase in the transfer of maternal lipids to the fetus [11].

Although ANGPTL3, 4, and 8 exert their function in a coordinated manner, the studies that have been performed so far on pregnancy have studied them individually. ANGPTL4 concentrations have been reported to increase progressively during normal gestation, with stable concentrations of glycerol and FFA [15], whereas ANGPTL3 and 8 concentrations decrease from the first to the third trimester [16,17]. In animal models, ANGPTL4 seems to play a role increasing maternal blood triglycerides during pregnancy [18], however, the role of the ANGPTL3-4-8 axis in the regulation of lipid metabolism during pregnancy has not been established in humans [15,16,17]. ANGPTL4 concentrations have been associated with an increased neonatal fat mass in pregnant women with gestational diabetes [19], and experimental models have been related to maternal blood triglyceride concentrations [18]. ANGPTL8 has been associated with the development of GDM [20] and offspring adiposity [16], whereas the ANGPTL3 concentrations were similar in pregnancies with and without pre-eclampsia [17], and they have not been studied in GDM. The placenta expresses ANGPTL3, ANGPTL4, and to a lesser extent, ANGPTL8, which appears to play a key role in implantation and angiogenesis [4,21], and it may play a role in fetal growth [7].

In this study, we propose to analyze the ANGPTL3-4-8 axis in normal and in GDM pregnancies. We assess the dynamics of free circulating ANGTPL3, 4, and 8 after the administration of a standard mixed meal, as described in the methods section. We will also evaluate the expressions of *ANGPTL3*, *4*, and *8* in the placenta and their relationship with newborn clinical parameters and the expressions of the proteins involved in placental lipid transport.

## 2. Results

### 2.1. ANGPTL Secretion Pattern after Meal Test

To assess the behavior of serum ANGPTL3, 4, and 8 at fasting and after feeding, we determined their concentrations at fasting and at 15′, 30′, 60′, 120′, and 180′ after the ingestion of a mixed meal. Simultaneously, the blood glucose, triglycerides, free fatty acids, and insulin concentrations were also determined. Table 1 shows the characteristics of the cohort of pregnant women included in the meal test study. The control and GDM groups were comparable, except for the pre-pregnancy BMI and insulin concentrations, which were higher in the group of women with GDM.

In the two-way ANOVA analysis, we observed differences between the ANGPTL4 response in normal gestations and gestations complicated by GDM (*p* < 0.05). Although the inhibition of ANGPTL3 levels was lower in the GDM group, this difference was not significant (*p* = 0.055). No difference was observed in the ANGPTL8 response (*p* = 0.837) (see Appendix A). 

Serum glucose, insulin, free fatty acids, and triglyceride curves are shown in Appendix A. The four analytical variables were comparable between the control and the GDM groups regarding their response to a meal intake test.

When we analyzed the dynamics of the ANGPTLs in each group separately, we observed that in the control population without GDM, the incremental area of ANGPTL3 was significantly lower than that of ANGPTL4, and this difference was not observed in the GDM population. The relationship between the ANGPTL8 levels with the other ANGPTLs was unchanged. These findings suggest dysregulation of the ANGPTL3-4 axis (see Figure 1).

### 2.2. Placental Expression and Location-Dependent Changes in Healthy Control Pregnancy and GDM

Next, we assessed the expressions of *ANGPTL3, 4*, and *8*, *LPL*, and the proteins related to lipid transport and metabolism in the placenta of healthy control women. The differences in gene expression have been described based on the placental localization. Given that a greater exposure to the maternal environment would be expected in the placental tissue closest to the uterine wall, that is, the maternal side of the placenta, we analyze whether the location could be associated with differences in gene expression. The clinical characteristics of the cohort included in this study are shown in Table 2. Since we observed differences in the pregestational BMI between the two groups, we wanted to determine its influence on the expressions of *ANGPTL3, 4,* and *8*. To do so, we divided the women according to their BMI (BMIs less than 25 and BMIs equal to or greater than 25), and we compared the expressions of *ANGPTL3, 4*, and *8* between the two groups. We found no differences between the two groups when both the groups were analyzed, nor when the control and GDM group were analyzed separately, except for the expression of *ANGPTL4* on the fetal side of the placenta in the GDM group. The expression value of *ANGPTL4* in the group with BMIs < 25 was higher than it was in the group with BMIs ≥ 25 (74.25 (56.0–91.96) vs. 39.73 (27.28–55.03); *p*-value: 0.027).

No differences were observed in *ANGPTL4* and *8* expressions between the maternal and fetal sides of the placenta. The *ANGPTL3* expression was found to be downregulated on the fetal side compared to that on the maternal side. We also observed a downregulation of *LPL* expression on the fetal side of the placenta. We did not observe any differences in the other genes that we studied (see Figure 2A).

We also examined the relationship between the expression of the different ANGPTLs at the same location and between the different locations. *ANGPTL4* and *ANGPTL8* expressions on the maternal side versus the fetal side of the placenta were positively related to each other (r = 0.724, *p* = 0.002; and r = 0.714, *p* = 0.003, respectively). No association was observed in the expression of *ANGPTL3* between the maternal and fetal sides of the placenta (see Appendix A).

Next, we performed the same analysis on the placentas from women with GDM, and we observed that the expressions of *ANGPTL3* and *FABP1* were significantly upregulated on the fetal side compared to the maternal side of the placenta. No other differences were observed (see Figure 2A). When we assessed the relationship between the ANGPTLs in this group of women, in addition to confirming the same positive relationship between the *ANGPTL4* and *ANGPTL8* expressions on both sides of the placenta (r = 0.684; *p* = 0.002 and r = 0.503; *p* = 0.047, respectively), we observed that *ANGPTL3* expression on the maternal side of the placenta was positively associated with *ANGPTL4* expression on both sides of the placenta (r = 0.832; *p* < 0.001 on the maternal side and r = 0.783; *p* = 0.003, on the fetal side). Furthermore, the *ANGPTL4* expression and *ANGPTL8* expression on the maternal side of the placenta were also associated (r = 0.614; *p* = 0.007) (see Appendix A).

### 2.3. Differences in ANGPTL3, 4, and 8, LPL, and Proteins Involved in Lipid Transport on Both Sides of the Placenta According to Glucose Tolerance Status

Next, we compared the placental gene expression in the GDM and control pregnancies. In the GDM women, a downregulation of *ANGPTL3* gene expression on the maternal side of the placenta was observed (see Figure 2B). No differences were observed between the two groups in the expression of the *ANGPTL4* and *8* genes, nor were differences observed in the expression of the *LPL* gene or the genes involved in the expression of the lipid transport-linked proteins studied, except for *FABP1*, which was overexpressed on the maternal side of the placenta of women with GDM compared with that of the controls (*p* < 0.01) (Appendix A).

### 2.4. ANGPTL3, 4, and 8 and Maternal and Offspring Characteristics

ANGPTLs have been associated with obesity and implicated in the regulation of lipid and glucose metabolism. For this reason, we examined whether the mRNA expression of the different ANGPTLs studied on both sides of the placenta showed associations with the clinical maternal characteristics, fetal metabolic parameters, and birth measurements of length and weight.

In the control group, no association was found with the maternal pregestational BMI, gestational weight gain, or gestational week at delivery, whereas *ANGPTL4* on the maternal side was negatively associated with the placental weight, but it was positively associated with the birth length (see Figure 3A).

As in the control group, no relationship was observed in the GDM group in the expression of the ANGPTLs and maternal parameters. By contrast, the expressions of all three ANGPTLs on the maternal side of the placenta were positively associated with birth weight, and in the case of *ANGPTL3* and *ANGPTL8*, they were also associated with placental size. The association between *ANGPTL4* on the fetal side of the placenta and birth length remained significant. Along the same lines, cord glycemia was associated with *ANGPTL4* on the maternal side, and cord blood insulin was associated negatively with *ANGPTL8* expression on both sides of the placenta (see Figure 3B).

### 2.5. Association of mRNA Gene Expression of ANGPTL3, 4, and 8, with mRNA Expression of Genes Involved in Placental Lipid Transport According to Glucose Tolerance Status

Finally, we also wanted to evaluate the relationship between the gene expression of proteins related to lipid transport and metabolism and the ANGPTLs.

*ANGPTL4* expression was positively associated with *CPT1A*, *CPT1B*, and *SLC27A4*, and with *FABP1*, and *FABP3* on the fetal side of the placenta. *ANGPTL3* on the fetal side of the placenta was positively associated with the expressions of *SLC27A2*, *FABP1*, and *CPT1A* only on the maternal side. No association was observed between any of the genes assessed and *ANGPTL3* on the maternal side of the placenta. *ANGPTL8* was positively associated with *LPL, SCL27A1*, and *SCL27A4* on the fetal side of the placenta and with *FABP3* on both sides of it (see Figure 4A).

When the same analysis was performed on gestations complicated with GDM, some differences were observed. *ANGPTL4*, in addition to the associations observed in normal gestation, showed a positive relationship with *FABP3* and *SLC27A4* on the maternal side of the placenta and with *LPL* on the fetal side of it. The associations of ANGPTL8 were maintained, except with *LPL*, which was lost. By contrast, a change in the association pattern of *ANGPTL3* was observed. The associations observed with fetal ANGPTL3 were lost. Instead, maternal *ANGPTL3* showed a positive association with *CPT1A* and *CPT1B* expressions on the maternal side and with maternal *SLC27A4* (see Figure 4B).

## 3. Discussion

This study provides us with new data on the behavior of the ANGPTL3-4-8 axis during normal pregnancies (controls) and pregnancies complicated by GDM. We show that GDM modifies the relationship between the ANGPTL3 and ANGPTL4 concentrations after a mixed meal in the third trimester of pregnancy. At the time of delivery, we revealed a glucose tolerance-dependent expression pattern of *ANGPTL3* in the placenta. In addition, we show how GDM modifies the relationship of the placental expressions of *ANGPTL3, 4*, and *8* with clinical parameters and with genes involved in lipid transport. The positive relationships between the *ANGPTL3, 4* and *8* expressions and birth weight in the GDM group suggests a dysregulation of this axis in GDM, which could be related to perinatal outcomes. 

It has been proposed that the ANGPTLs may play a role in the regulation of gestational lipid metabolism, and that the system may be dysregulated in pregnancies complicated by GDM [22,23]. However, as far as we know, they have not been investigated together before in pregnancies complicated by GDM. In this small cohort of pregnant women with and without GDM, we observed that the ANGPTL3 and ANGPTL4 responses were unbalanced, with no elevation of the ANGPTL4 levels and less inhibition of ANGPTL3 in women with GDM. ANGPTL proteins have been studied after a meal test, with inconclusive results. Similar to what we found, Chen et al. observed that free ANGPTL3 concentrations did not change after a mixed meal challenge in normal subjects [9]. It is of note that this test has not been performed on pregnant women. In contrast, Schmid et al. reported that the ANGPTL3 concentrations decreased and those of ANGPTL4 increased after an oral lipid test in healthy individuals, while no changes in the ANGPTLs concentrations were observed after an oral glucose tolerance test [24]. It is difficult to draw conclusions from such a subtle alteration, especially if we take into account that circulating ANGPTL4 levels seem to play a minor role in the regulation of lipid metabolism, and that its effect on the circulating triglyceride levels seems to be due to its local tissue action [9,25]. Moreover, the inhibitory action of ANGPTL3 on LPL is exerted mainly through binding with ANGPTL8, thereby forming complexes. Free ANGPTL3 seems to have a minor role in the regulation of plasma lipids. Furthermore, we did not observe any change in the ANGPTL8 levels, even though its expression and secretion are known to be stimulated by feeding. This finding may be due to the fact that the method used only determines the free ANGPTL3, 4, and 8 levels. Additionally, it seems that ANGPTL8 that is secreted after ingestion is found in the form of complexes linked to ANGPTL3, which potentiates its inhibitory action on LPL. Finally, at the end of gestation, it is worth noting that the LPL activity is decreased due to the presence of insulin resistance [26].

Although, ANGPTL3, 4, and 8 are mainly expressed and secreted by the liver and adipose tissue, they are also expressed in the placenta, and they may play a role in modulating the passage of nutrients across the placenta. Although most studies do not differentiate between the two sides of the placenta, it has been observed that there may be differences in the gene expressions depending on the anatomical location of the tissue [27,28]. The data from our cohort support that the anatomical location is associated with differences in the expression of some genes and that this expression may also be conditioned by the metabolic status. In line with the results of Qiao et al. in adipose tissue [18], obesity results in a lower expression of *ANGPTL4* on the fetal side of the placenta. This finding, along with the lower expression of *ANGPTL3* also on the fetal side, may enhance *LPL* and/or endothelial lipase, favoring placental free fatty acid uptake.

A recent study on Hispanic GDM women with and without macrosomias shows that *ANGPTL3* mRNA is overexpressed in the placentas of women with GDM and macrosomias, with no differences in the expressions of *ANGPTL4* and *ANGPTL8*. Of note, in this same study, the *ANGPTL3* expression in the placentas from women with GDM and a non-macrosomic child was decreased compared to that of the controls [7]. Given that in our study the mean weight of the newborns was similar in both of the groups and that there was only one case of macrosomia, the *ANGPTL3* downregulation observed on the maternal side of placentas of women with GDM would be in agreement with the aforementioned study. It should be noted that endothelial lipase is also a target of ANGPTL3 [29], and it has been implicated in fetal growth in pregnancies complicated by diabetes [30,31] and in intrauterine growth retardation [32]. Therefore, a lower expression of *ANGPTL3*, specifically on the maternal side of the placenta in the GDM group, could be accompanied by increased activity of this enzyme, facilitating the transplacental passage of nutrients. Additionally, the lower expression of *FABP1* in pregnant women with GDM would be in line with previous studies that have been performed on women with obesity [33,34].

An intriguing aspect is the positive relationship between the *LPL* and *ANGPTL4* expressions in women with GDM. We do not know whether this relationship is due to protective or detrimental effects. It has been reported that ANGPTL4 plays an important role in limiting LPL activity on macrophages and cardiomyocytes, reducing lipid uptake, and protecting them against oxidative stress [35,36]. Additionally, we know that LPL activity is regulated at the post-translational level and that the inhibitory action exerted by ANGPTL4 is tissue specific and depends on its binding to ANGPTL8, forming complexes [25]. In the latter situation, if the behavior in the placenta is similar to that in adipose tissue, we would expect to find more LPL activity in the presence of the ANGPTL4 and 8 complexes, and this could explain the association between ANGPTL4 and birth weight in women with GDM.

Another mechanism that could justify the association between ANGPTLs expression and birth weight is an increased passage of nutrients due to a larger exchange surface. It is well known that the placental growth in pregnancies complicated with diabetes responds to the enhanced fetal demand and is achieved mostly by angiogenesis, and ANGPTLs have been involved in the regulation of angiogenesis [4,21,37]. Additionally, the association observed between the expression of *ANGPTL4* on the fetal side of the placenta and birth length in both the healthy control and GDM pregnancies suggests that ANGPTL4 regulates nutrient transfer and determines the birth size.

Inflammation, another aspect that we have not analyzed in this study, could be related to placental expression of the ANGPTL3-4-8 axis. Although there is only a little knowledge about the relationship of ANGPTLs and inflammation in placentas at term, given that ANGPTLs have been related to inflammation [38] and that the placentas from women with GDM appear to be more inflamed [39], the observed changes in ANGPTL3 expression could also be partly due to this factor. In fact, the evidence in other tissues and situations has shown that ANGPTL3 seems to be related more to proinflammatory activity, whereas ANGPTL4 and ANGPTL8 may show protective or proinflammatory activities depending on the circumstances [38].

As this was an observational study, causality cannot be inferred. Another aspect to consider is that we only determined the levels of free ANGPTL3, ANGPTL4, and ANGPTL8, whose role in the regulation of lipid metabolism could be more limited. Additionally, despite the information provided on mRNA expression of the three ANGPTLs in the placenta, the determination of LPL activity or the measurement of protein expression would have provided valuable additional information. We feel that our study has several strengths, such as addressing the ANGPTL3-4-8 axis globally in normal gestation, as well as in those complicated with GDM. We evaluated the expression of the ANGPTL3-4-8 axis in a large cohort of placentas from women with and without GDM, allowing us to establish relationships between the lipid transporters and neonatal anthropometric parameters, which may be a starting point for further studies.

For the first time, we demonstrated that the ANGPTL3-4-8 axis appears to be dysregulated in pregnancies with gestational diabetes, both at the circulating level after ingestion and at the level of placental expression. The association observed between placental ANGPTLs expression and birth weight and length in GDM, but not in the controls, suggests that they may play a role in regulating the transplacental passage of nutrients.

## 4. Methods

### 4.1. Study Population

This was an observational study carried out at the Joan XXIII University Hospital of Tarragona, with the aim of evaluating the dysregulation of the ANGPTL3-4-8 axis in gestation complicated by gestational diabetes.

Pregnant women with and without gestational diabetes were invited to participate in this prospective observational study. This study has two parts: one part focused on the assessment of ANGPTL dynamics after feeding, and a second part focused on the evaluation of ANGPTL mRNA expression in the placenta. For this purpose, we included two independent cohorts of pregnant women with normal glucose tolerance (healthy controls) and GDM. The first cohort consisted of eighteen pregnant women, eight of them were in the healthy control group (HC) and ten women with GDM, who underwent a meal test to study the dynamics of the ANGPTL3-4-8 axis during gestation. In the second cohort, consisting of 39 pregnant women (19 in the HC group and 20 in the GDM group), the gene expressions of ANGPTL3, ANGPTL4 and ANGPTL8 in the placenta were studied. Four women with GDM participated in both parts of the study.

All of the women were followed at the Obstetric Department of the Joan XXIII University Hospital. All of the participants were screened for GDM at 24–28 weeks of gestation using a two-step approximation according to the standards of the Spanish Group on Diabetes and Pregnancy guidelines, which followed the National Data Group Criteria [40,41]. Pregnant women with either pre-existing Type 1 or Type 2 diabetes, inflammatory or chronic diseases, or those who were taking drugs that are known to affect carbohydrate metabolism, were excluded from the study cohort.

The study was performed according to the tenets of the Helsinki Declaration. Ethical approval was obtained by the Joan XXIII University Hospital Ethics Board (243C/2016), and all of the participants signed a written informed consent before entering the study.

Gestational age was confirmed by a routine ultrasonographic examination, which was scheduled before week 20 of gestation. Maternal pre-pregnancy weight and height was annotated in a self-report at the first prenatal visit. Pre-pregnancy body mass index (BMI) and gestational weight gain (GWG) were calculated using the following formulas: pre-pregnancy BMI = pre-pregnancy weight (kg)/(height (m))^2^ and GWG = final weight − pre-pregnancy weight. The gestational timing of delivery was based mainly on obstetric indications. Neonatal birth weight and length were annotated within 48 h of parturition.

### 4.2. Meal Test

Sera were obtained from pregnant women during the third trimester of pregnancy (32–35 weeks). To study the fasting and postprandial conditions, sera were obtained after overnight fasting and 15′, 30′, 60′, 120′, and 180′ following a liquid meal test that contained 400 kcal (20 g fat, of which 1% was saturated, 72% was monounsaturated and 20% was polyunsaturated; 47 g of carbohydrate; 8 g of protein) (Renilon 4.0^®^; Nutricia, 28043 Madrid, Spain). All of the samples were stored at −80 °C.

### 4.3. Cord Blood and Placental Collection and Processing

Umbilical cord blood was obtained immediately after the delivery. The serum was then separated by centrifugation, divided into aliquots, and stored at −80 °C until further determination. Full-term placentas (37–39 weeks gestation) were collected after delivery and processed without delay under sterile conditions. After the separation of the amniotic membrane and the chorion, villous tissue surface sections from the maternal and fetal sides of the placenta were gently washed in PBS. Next, the villus parenchyma sections were obtained by dissecting a square-shaped segment of between 1.5 and 3 cm through the entire ≈2.5 cm thickness of the placental disk (≈5 cm away from site of cord insertion), and then splitting it into two equal parts: the maternal and fetal parts. The tissues were frozen in liquid N_2_ and stored at −80 °C. Total RNA from the tissues was isolated using the RNeasy Mini kit (Qiagen, Valencia, CA, USA), and its quality was assessed by the 260/280 nm optical density ratio. RNA was transcribed into cDNA with random primers using a dNTP Mix (100 mM), MultiScribe Reverse Transcriptase (50 U/μL) and RNase Inhibitors using the High-Capacity cDNA Reverse Transcription Kit (Applied Biosystems, Foster City, CA, USA). Gene expression was evaluated by quantitative reverse transcription in real-time PCR (RT-qPCR) on a 7900HT Fast Real-Time PCR System (Applied Biosystems) using a predesigned TaqMan Low Density Array (Applied Biosystems). The genes that were evaluated were: *ANGPTL3* (Hs00205581_m1), *ANGPTL4* (Hs01101123_g1), *ANGPTL8* (Hs00218820_m1), *LPL* (Hs00173425_m1), *CD36* (Hs00354519_m1), *FATP1/SLC27A1* (Hs01587911_m1), *FATP2/SLC27A2* (Hs00186324_m1), *FATP4/SLC27A4* (Hs00192700_m1), *FATP6/SLC27A6* (Hs00204034_m1), *FABP1* (Hs00155026_m1), *FABP3* (Hs00997360_m1), *CPT1A* (Hs00912671_m1), and *CPT1B* (Hs00189258_m1). Gene expression values were calculated using the comparative Ct method (2^−ΔΔCt^) and normalized to the expression of the housekeeping gene *18S* (Hs03928985_g1).

### 4.4. Laboratory Measurements

Serum glucose, cholesterol, and triglyceride levels were determined using ADVIA 1800 and 2400 (Siemens AG, Munich, Germany) autoanalyzer platforms following standard enzymatic methods. LDL cholesterol was calculated using the Friedewald formula. Serum insulin was determined in an immunoassay in an ADVIA Centaur System (Siemens AG, Munich, Germany). This assay shows a cross-reactivity of 0.1% to intact human proinsulin and the primary circulating split form des 31, 32 proinsulin.

### 4.5. ANGTPL3, 4 and 8 Determinations

The serum ANGPTL3, 4, and 8 levels were measured by ELISA (MyBioSource Inc., San Diego, CA, USA) following the manufacturer’s instructions. Concretely, the Human Angiopoietin-Like Protein 3 ELISA Kit (Ref. MBS008458), the Human Angiopoietin-Like Protein 4 ELISA Kit (Ref. MBS3802882), and the Human Angiopoietin-Like Protein 8 ELISA Kit (Ref. MBS2021947) were used.

### 4.6. Data Calculations

The insulinogenic index was obtained using the equation (Insulin [mU/mL] 30′-0′)/(Glucose [mg/dL] 30′-0′). The Area Under the time–concentration Curve (AUC) was calculated using the trapezoidal rule. The feeding effect on ANGPTLs was calculated as a fold increase in the fasting values (FC = ([follow up-baseline]/baseline) + 1). Insulin resistance was estimated using the oral glucose insulin sensitivity index [42].

### 4.7. Statistics

All of the data were tested for normality using the Shapiro–Wilk test. For the meal test samples, the values are reported as mean ± S.E.M. The significance of meal effects on the circulating ANGPTL3, 4, and 8 levels was assessed using repeated measures ANOVA with a Bonferroni post hoc test (*p*-values refer to differences among ANGPTLs curves). Area Under the Curve (AUC) differences were assessed by the Kruskal–Wallis test, followed by Dunnett’s correction. Gene expression data are reported as log_2_(fold change), and the differences were assessed by performing a two-way ANOVA multiple comparison test, followed by the Bonferroni correction. Correlations were calculated using Spearman’s correlation coefficient.

For the clinical and anthropometrical variables, the data are expressed as the median and 25th–75th percentiles (interquartile range (IQR)). A chi-squared test was used to compare the qualitative variables, and differences in the quantitative variables between the groups were determined using a non-parametric Mann–Whitney U test.

Statistical calculations and visualizations were performed using GraphPad software version 8.0 (GraphPad Software Inc., San Diego, CA, USA). For clinical, metabolic, and gene expression data, data integration SPSS software version 20.0 was used (IBM, Armonk, NY, USA). A *p*-value < 0.05 was considered to be statistically significant.

## Figures and Tables

**Figure 1 ijms-24-02486-f001:**
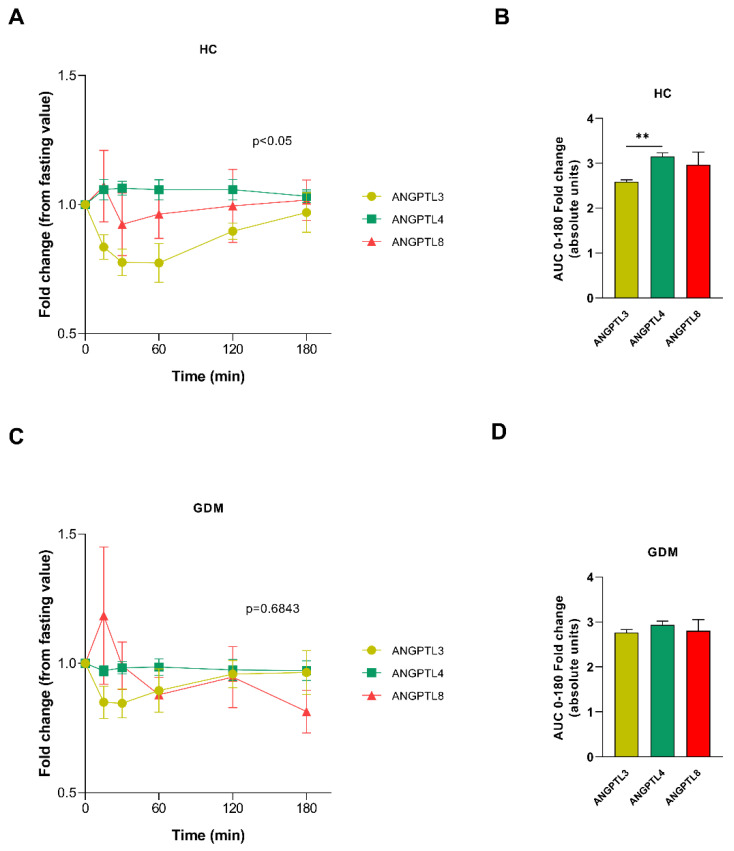
ANGPTL response to a meal test. The figure shows time curves of serum ANGPTL3, 4, and 8 in healthy control pregnant women (**A**) and in GDM women (**C**) during a meal test (fold increase over basal values). (**B**,**D**) show the Area Under the Curve values (AUC) of ANGPTL3, 4, and 8 in healthy control pregnant women and GDM women, respectively. Data are mean ± S.E.M. AUC differences were assessed by Kruskal–Wallis test, followed by Dunnett’s correction, and the time curves were compared using repeated measures ANOVA with Bonferroni post hoc test (*p*-values refer to differences among ANGPTLs curves). ** *p* < 0.01. HC: healthy control; GDM: gestational diabetes; ANGPTL3: Angiopoietin-Like Protein 3; ANGPTL4: Human Angiopoietin-Like Protein 4; ANGPTL8: Human Angiopoietin-Like Protein 8.

**Figure 2 ijms-24-02486-f002:**
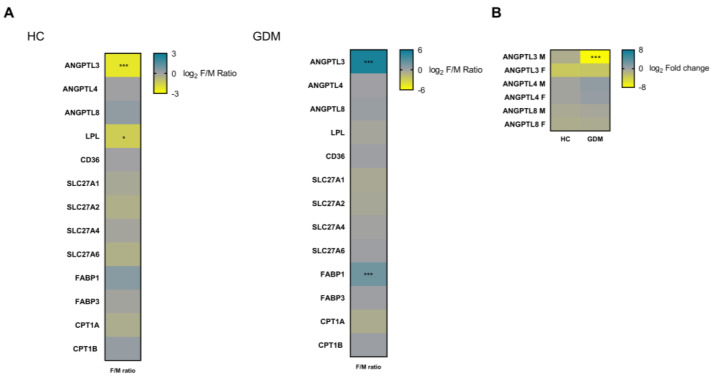
Comparison of gene expression in distinct parts of the placenta. (**A**) Heatmap comparing ANGPTLs, lipoprotein lipase and lipid transport gene expressions as a ratio between the fetal and maternal side of the placenta (**left panel**) in healthy control women (*n* = 19 in each of them) and (**right panel**) in the gestational diabetes group (*n* = 20). (**B**) Heatmap comparing *ANGTPL3, 4*, and *8* expressions on both sides of the placenta according to glucose tolerance status (HC vs. GDM). Data are represented as log2 (F/M ratio) (**A**) and log2 (fold change) (**B**). Differences assessed by two-way ANOVA multiple comparison test, followed by Bonferroni correction. Data are normalized to intraindividual gene expression on the maternal side of the placenta (**A**) and to the mean of the control group for each gene and side (**B**) and are shown as means. M and F indicate gene expressions on the maternal and fetal sides of the placenta, respectively. * *p* < 0.05; *** *p* < 0.001. HC: healthy control; GDM: gestational diabetes. ANGPTL3: Angiopoietin-Like Protein 3; ANGPTL4: Human Angiopoietin-Like Protein 4; ANGPTL8: Human Angiopoietin-Like Protein 8; LPL: Lipoprotein Lipase, CD36: Cluster of Differentiation 36; SLC27A1, SLC27A2, SLC27A4, and SLC27A6: Solute Carrier Family 27 Members 1, 2, 4, and 6, respectively; FABP1 and FABP3: Fatty Acid Binding Proteins 1 and 3, respectively; CPT1A and CPT1B: Carnitine Palmitoyltransferase 1A and 1B, respectively.

**Figure 3 ijms-24-02486-f003:**
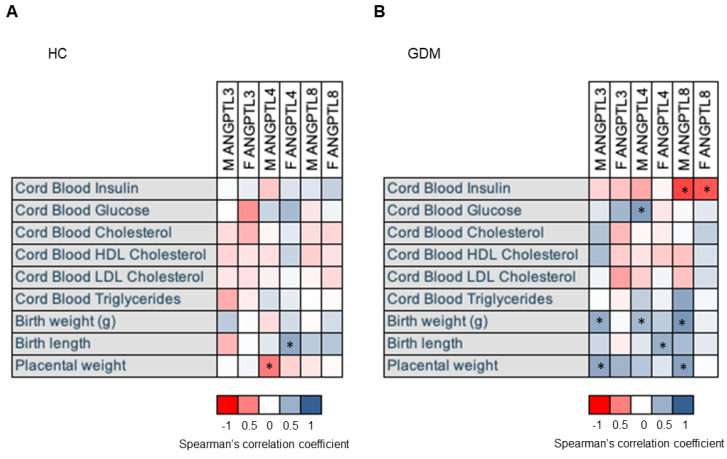
ANGPTL expression is associated with offspring parameters in the GDM group. Heatmap of the correlation coefficients observed between *ANGPTL3, 4*, and *8* placental expressions and neonatal clinical variables in control (**A**) and GDM (**B**) mothers. Gene expression data are expressed as fold change. Correlations were calculated using Spearman’s correlation coefficient. * *p* < 0.05. HC: healthy control; GDM: gestational diabetes mellitus; M before the name of any gene expression indicates expression on the maternal side of the placenta; F before the name of any gene expression indicates expression on the fetal side of the placenta. ANGPTL3: Angiopoietin-Like Protein 3; ANGPTL4: Human Angiopoietin-Like Protein 4; ANGPTL8: Human Angiopoietin-Like Protein 8.

**Figure 4 ijms-24-02486-f004:**
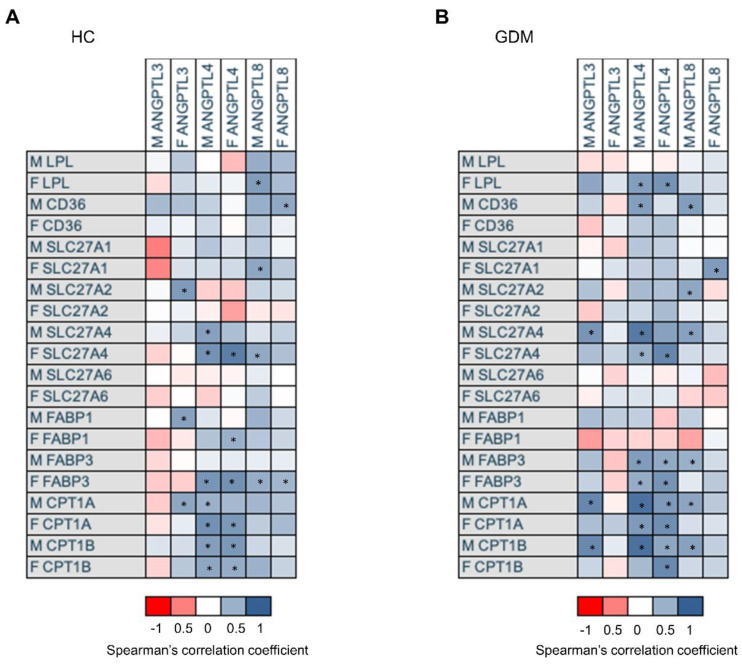
The relationship between the expressions of ANGPTL3, 4 and 8 and the expressions of genes involved in placental lipid transport is modified in gestational diabetes. Heatmap of the correlation coefficients observed between *ANGPTL3, 4*, and *8* placental expressions and the expressions of genes involved in lipid transport in control (**A**) and GDM (**B**) mothers. Gene expression data are expressed as fold change. Correlations were calculated using Spearman’s correlation coefficient. * *p* < 0.05. HC: healthy control; GDM: gestational diabetes mellitus; M before the name of any gene expression indicates expression on the maternal side of the placenta; F before the name of any gene expression indicates expression on the fetal side of the placenta. ANGPTL3: Angiopoietin Like Protein 3; ANGPTL4: Human Angiopoietin-Like Protein 4; ANGPTL8: Human Angiopoietin-Like Protein 8; LPL: Lipoprotein Lipase, CD36: Cluster of Differentiation 36; SLC27A1, SLC27A2, SLC27A4, and SLC27A6: Solute Carrier Family 27 Members 1, 2, 4, and 6, respectively; FABP1 and FABP3: Fatty Acid Binding Proteins 1 and 3, respectively; CPT1A and CPT1B: Carnitine Palmitoyltransferase 1A and 1B, respectively.

**Table 1 ijms-24-02486-t001:** Characteristics of pregnant women included in the meal test study.

	Healthy Control *n* = 8	GDM *n* = 10	*p*-Value
Age (years)	32.0 (31.3–35.3)	34.0 (26.0–36.0)	0.968
BMI (kg/m^2^)	21.9 (18.4–29.8)	28.0 (26.3–36.3)	0.041
GWG (kg)	11.3 (9.4–13.6)	6.7 (0.5–9.6)	0.062
Gestational Week at testing	32.0 (32.3–33.0)	32.0 (30–34.0)	0.968
Glucose (mmol/L)	3.8 (3.7–4.4)	4.2 (4.1–4.5)	0.091
Triglycerides (mmol/L)	1.9 (1.5–2.7)	2.4 (2.1–3.1)	0.177
FFAs (nmol/mL)	77.4 (53.0–98.8)	76.7 (59.5–1050)	1
Insulin (pmol/L)	50.9 (45.2–91.1)	122.0 (68.2–177.9)	0.015
ANGPTL3 (ng/mL)	202 (129–224)	240 (164–340)	0.167
ANGPTL4 (ng/mL)	6.9 (5.9–7.9)	7.6 (6.2–8.2)	0.091
ANGPTL8 (pg/mL)	535 (417–1059)	531 (437–735)	0.875
Insulinogenic index	0.6709 (0.2953–1.739)	0.8150 (0.6143–1.523)	0.6058
OGIS Index (mL/min/m^2^)	550.5 (484.3–622.8)	482.0 (446–520.5)	0.0745

GDM: gestational diabetes Mellitus; BMI: body mass index; GWG: gestational weight gain; FFAs: Free Fatty Acids; ANGPTL3: Angiopoietin Like Protein 3; ANGPTL4: Human Angiopoietin-Like Protein 4; ANGPTL8: Human Angiopoietin-Like Protein 8.

**Table 2 ijms-24-02486-t002:** Characteristics of the cohort included in the placental expression analysis.

	Healthy Control *n* = 19	GDM *n* = 20	*p*-Value
Age (years)	35.0 (31.0–40.0)	35.0 (32.5–38.0)	0.968
BMI (kg/m^2^)	22.7 (21.03–31.1)	27.6 (22.6–29.4)	0.041
GWG (kg)	10.0 (5.5–14.0)	8.1 (5.5–10.7)	0.062
Gestational Week at delivery	39.0 (38.0–40.0)	39.0 (38.0–40.0)	0.968
Birth weight (g)	3388 (3080–3590)	3330 (3125–3537)	0.091
Birth length (cm)	49.0 (48.0–50.0)	49.5 (48.1–51.0)	0.177
Placental weight (g)	600 (540–660)	615 (525–700)	1
Cord blood glucose (mmol/L)	4.3 (3.6–4.9)	4.2 (4.0–5.1)	0.723
Cord blood Cholesterol (mmol/L)	1.6 (1.5–2.1)	1.9 (1.6–2.0)	0.415
Cord blood Cholesterol HDL (mmol/L)	0.70 (0.65–0.75)	0.73 (0.62–0.83)	0.884
Cord blood Cholesterol LDL (mmol/L)	0.98 (0.65–0.96)	0.91 (0.67–0.96)	0.415
Cord blood Triglycerides (mmol/L)	0.40 (0.28–0.58)	0.61 (0.36–0.69)	0.072
Cord blood Insulin (pmol/L)	20.8 (10.8–48.1)	26.6 (7.9–62.4)	0.947

GDM: gestational diabetes mellitus; BMI: body mass index; GWG: gestational weight gain.

## Data Availability

Data are available from authors upon reasonable request.

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
