# Peer review of "The ANGPTL3-4-8 Axis in Normal Gestation and in Gestational Diabetes, and Its Potential Involvement in Fetal Growth"

_ijms, 2023, doi:10.3390/ijms24032486_

Round 1
Reviewer 1 Report
The ANGPTL 3, 4, and 8 axis in normal gestation and in gestational diabetes, and its potential involvement in fetal growth. By Sergiy KLID et al.
This paper is suitable for publication. The area of interest is towards the metabolic changes and gene expression in pregnancy, normal and GDM. From the results, no direct impact in clinical usefulness is conceivable to date. “Four women with GDM participated in both parts of the study” in the Study population paragraph (4.1.) should be clarified, the studied subjects are quite limited, and the statement may be not clear to the reader.
Author Response
The ANGPTL 3, 4, and 8 axis in normal gestation and in gestational diabetes, and its potential involvement in fetal growth. By Sergiy KLID et al.
Point 1: This paper is suitable for publication. The area of interest is towards the metabolic changes and gene expression in pregnancy, normal and GDM. From the results, no direct impact in clinical usefulness is conceivable to date. “Four women with GDM participated in both parts of the study” in the Study population paragraph (4.1.) should be clarified, the studied subjects are quite limited, and the statement may be not clear to the reader.
Response 1: Thank you for your comment. We have rewritten the paragraph describing the two cohorts to make it easier to understand.

Reviewer 2 Report
Angiopoietin-like (ANGPTL) family, which is composed of eight members, mainly express in and secreted liver adipose tissue and placenta. It well known that ANGPTL inhibit lipoprotein lipase (LPL) enzyme activity selectively and plays an essential role in metabolic homeostasis maintenance. ANGPTL3, ANGPTL4, and ANGPTL8 are important members of ANGPTL family and regulate plasma triglyceride levels, as well as stem cell renewal, angiogenesis, and inflammation. The relationship between ANGPTL3, ANGPTL4, and ANGPTL8 and gestational diabetes is still unknown. In this study, authors found sharply decreased ANGPTL4 in gestational diabetes group (GDM) after meal, compare to the ANGPTL4 expression in normal group. The data revealed ANGPTL3 expression in placental tissue is glucose tolerance dependent. Furthermore, the authors also found that the expression of ANGPTL3, 4, and 8 was related to birth weight and placental weight in the GDM group. It suggests these genes play a key role in nutrients transactions in placenta. This study is interesting; however, some concerns below should be addressed.
1. The authors found different gene expression in distinct parts of the placenta. The method of placenta distinct part identification should be described clearly. Why choose these parts of placenta?
2. ANGPTL3, ANGPTL4, and ANGPTL8 are associated with inflammation, what about the inflammatory factor expression in placenta form GDM patients?
3. ANGPTL family regulates the lipid metabolism, what about the ANGPTL3, ANGPTL4, and ANGPTL8 expression in patients in different BMI or LDL?
4. Placenta growth factor (Plgf) is also associated with GDM. Lack of Plgf may lead to gestational diabetes, and glucose tolerance. Moreover, Plgf not only expresses in placental, but also expresses in pancreatic islet. What about the relationship between Plgf and ANGPTL3 family? Reference (PMID: 35546452) should be fully discussed.
Author Response
Angiopoietin-like (ANGPTL) family, which is composed of eight members, mainly express in and secreted liver adipose tissue and placenta. It well known that ANGPTL inhibit lipoprotein lipase (LPL) enzyme activity selectively and plays an essential role in metabolic homeostasis maintenance. ANGPTL3, ANGPTL4, and ANGPTL8 are important members of ANGPTL family and regulate plasma triglyceride levels, as well as stem cell renewal, angiogenesis, and inflammation. The relationship between ANGPTL3, ANGPTL4, and ANGPTL8 and gestational diabetes is still unknown. In this study, authors found sharply decreased ANGPTL4 in gestational diabetes group (GDM) after meal, compared to the ANGPTL4 expression in normal group. The data revealed ANGPTL3 expression in placental tissue is glucose tolerance dependent. Furthermore, the authors also found that the expression of ANGPTL3, 4, and 8 was related to birth weight and placental weight in the GDM group. It suggests these genes play a key role in nutrients transactions in placenta. This study is interesting; however, some concerns below should be addressed.
Point 1: The authors found different gene expression in distinct parts of the placenta. The method of placenta distinct part identification should be described clearly. Why choose these parts of placenta?
Response 1: We agree with the reviewer that the explanation of the method we used to process and establish the two placental sections was inaccurate. We have expanded the information to make it easier to understand.
Previous studies have already determined differences in the expression of placental genes depending on their location. Given that the placenta acts as a barrier between the mother and the fetus, and is subject to metabolic and hormonal stimuli, we were interested in knowing whether proximity to the maternal or fetal environment could condition differences in the expression of these genes.
We have introduced a clarifying sentence in the first paragraph of point 2.2 of the results section
Point 2: ANGPTL3, ANGPTL4, and ANGPTL8 are associated with inflammation, what about the inflammatory factor expression in placenta form GDM patients?
Response 2: An interesting observation. There is evidence linking ANGPTLs to inflammation and GDM has been associated with increased placental inflammation. However, little is known about the relationship of the ANGPTL3, 4, and 8 axis to inflammatory factors in the placenta at term, but the inflammatory response could be a determinant of the observed changes in placental expression.
We have introduced a paragraph in the discussion in which we comment on the possible involvement of inflammation in the regulation of the expression of placental ANGPTLs.
Point 3: ANGPTL family regulates the lipid metabolism, what about the ANGPTL3, ANGPTL4, and ANGPTL8 expression in patients in different BMI or LDL?
Response 3: We agree that it would be interesting to analyze the expression of ANGPTL3, 4, and 8 as a function of the degree of obesity and maternal LDL. Unfortunately, we do not have the LDL values at the time of delivery of the women studied in this cohort, but we have performed the analysis according to the presence or absence of overweight/obesity.
We divided the women into two groups, BMI < 25 and BMI ≥ to 25, and compared the levels of ANGPTL3, 4, and 8 expression in placenta. We have not observed differences in expression between the two groups when the whole group was considered (see table), nor when the control and GDM group were analyzed separately, except for the expression of ANGPTL4 on the fetal side of the placenta in the GDM group. The expression of ANGPTL4 in the group with BMI < 25, was greater than in the group with BMI ≥ 25 (74.25 (56.0-91.96 vs 39.73 (27.28-55.03); P-value: 0.027).
Table. ANGPLT3, 4, and 8 expression levels according to BMI.
|
|
BMI < 25 |
BMI ≥ 25 |
P-value |
|
M ANGPTL3 |
0.08 (0.01-0.91) |
0.01 (0.00-0.359) |
0.161 |
|
F ANGPTL3 |
0.73 (0.14-2.90) |
0.80 (0.01-1.52) |
0.515 |
|
M ANGPTL4 |
38.75 (25.32-84.43) |
25.10 (15.65-78.06) |
0.208 |
|
F ANGPTL4 |
37.85 (21.86-69.86) |
34.02 (24.66-54.96) |
0.424 |
|
M ANGPTL8 |
0.006 (0.002-0.017) |
0.003 (0.001-0.009) |
0.309 |
|
F ANGPTL8 |
0.006 (0.002-0. 360) |
0.004 (0.003-0.010) |
0.452 |
A brief explanation has been included in the first paragraph of point 2.2, in the Results section.
Point 4: Placenta growth factor (Plgf) is also associated with GDM. Lack of Plgf may lead to gestational diabetes, and glucose tolerance. Moreover, Plgf not only expresses in placental, but also expresses in pancreatic islet. What about the relationship between Plgf and ANGPTL3 family? Reference (PMID: 35546452) should be fully discussed.
Response 4: Yes, we are aware tha PlGF has been associated with GDM and a lack of PlGF may lead to glucose intolerance. It would have been very interesting to have data on the expression of PlGF and FLT1 in placenta. Unfortunately, we do not have this data at this time.
In relation to the article suggested by the reviewer, we have read it carefully and found it very interesting, but we have not found any connection point that would allow us to make a full discussion.
